# The Effect of Phenyl Content on the Liquid Crystal-Based Organosilicone Elastomers with Mechanical Adaptability

**DOI:** 10.3390/polym14050903

**Published:** 2022-02-24

**Authors:** Zhe Liu, Hua Wang, Chuanjian Zhou

**Affiliations:** School of Materials Science and Engineering, Shandong University, Jinan 250061, China; 201820362@mail.sdu.edu.cn (Z.L.); hwang@sdu.edu.cn (H.W.)

**Keywords:** liquid crystal-based phenyl silicone rubber, phenyl content, the regulation of mechanical adaptability, damping performance

## Abstract

An elastomer with mechanical adaptability is a new kind of polymer material in which the increasing stress under continuous deformation is significantly inhibited in a large deformation area. Liquid crystal-based organosilicone elastomers, which can dissipate energy through reversible internal phase transition under external stimulation and have recoverable large deformation capacity, have drawn much interest as mechanical adaptability materials. However, there is no good way to control the mechanical adaptability at present. For this purpose, we prepared a new liquid crystal-based phenyl silicone rubber (LCMVPQ) using two-step click reactions and systematically explored the effect of phenyl content on its mechanical adaptability to achieve the regulation of mechanical adaptability. With an increase in phenyl content in the LCMVPQs, phenyl can hinder the rearrangement of the mesogenic units along the applied stress direction, which enables the adjustment of mechanical adaptability to meet the needs of different situations. In addition, the introduction of the liquid crystal phase impedes the internal friction of the molecular chain movement of the LCMVPQs and reduces the damping performance of silicone rubber. This research achieves the regulation of elastomers with mechanical adaptability and is expected to be applied in practical application fields.

## 1. Introduction

In the last decades, numerous strategies have been adopted to develop a new kind of elastomer with mechanical adaptability [1,2,3]. Mechanical adaptability is the ability to significantly suppress the increasing stress under continuous deformation in a large deformation region [4,5,6,7,8]. For widely used soft-connected systems, it is necessary not only to provide some mechanical strength under stress, but also to expand the stress platform as much as possible under harsh service conditions [9,10]. Liquid crystal elastomers [11,12,13,14], which can dissipate energy through reversible internal phase transition under external stimulation and have recoverable large deformation capacity [15,16,17], have become one of the best choices for mechanical adaptability materials [18,19]. Recently, White et al. explored stress-induced phase transitions in elastomers with appreciable liquid crystal content and compared these responses to LCEs in the polydomain orientation [1]. Ware’s group achieved the 3D-structure printing of thermal-responsive liquid crystal elastomers (LCEs) by controlling the molecular order for volumetric contractions or rapid, repetitive snap-through transitions [20].

Although liquid crystal elastomers have attracted extensive attention due to their excellent properties [7,21,22], few research groups have focused on their mechanical adaptability [23,24,25]. More importantly, there is no effective method to control the mechanical adaptability at present to meet the needs of different situations [26,27,28].

Here, we prepared a series of LCMVPQs with different phenyl content and their phenyl silicone rubber analogs and systematically explored the effect of phenyl content on their mechanical adaptability to achieve the regulation of mechanical adaptability. The structure of the LCMVPQs was confirmed by nuclear magnetic resonance hydrogen spectrum (^1^H NMR) and Fourier transform infrared spectroscopy (FT-IR). The thermal property was evaluated by differential scanning calorimetry (DSC) and thermogravimetric analysis (TGA). The effect of phenyl content on its mechanical adaptability was explored using a universal tensile tester and dynamic thermomechanical analysis (DMA). The results show that the steric hindrance of phenyl can hinder the rearrangement of the mesogenic units along the applied stress direction. This feature can be used to adjust the mechanical adaptability to meet the needs of different situations. Moreover, the presence of the liquid crystal phase hinders the internal friction between the molecular segments of LCMVPQ and reduces the damping property, which has not yet been reported. This study investigated the adjustment of the mechanical adaptability of silicone rubber, which is of great significance for the application of this key material.

## 2. Experimental Section

### 2.1. Materials

Divinyltetramethyldisiloxane (MM^vi^) and 2,4,6,8-tetramethyltetravinylcyclotetrasiloxane (D_4_^vi^) were obtained from Chenguang Chemical Co., Ltd. (Qufu, Shandong, China). Octamethylcyclotetrasiloxane (D_4_) and methylphenylcyclosiloxane (D_n_
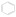
) were purchased from Empire Forever Chemical Co., Ltd. (Nanjing, China). 4-Methoxyphenyl 4-(3-Butenyloxy) benzoate (MBB) was obtained from Titan Scientific Co., Ltd. (Shanghai, China). 2,2-Dimethoxy-2-phenylacetophenone (DMPA) was obtained from Aladdin Chemical Corporation (Shanghai, China). Trifluoromethanesulfonic acid (≥98%, J&K Scientific, Beijing, China), Toluene (≥99.5%, Sinopharm Chemical Reagent Co., Ltd., Shanghai, China), dichloromethane (DCM, ≥99.5%, Sinopharm Chemical Reagent Co., Ltd., Shanghai, China). The nano-silica (H2000) was purchased from Wacker Chemie AG Corporation (Shanghai, China). Silicone oil modified by thiol group (SHSO) and tetramethylammonium hydroxide silicon alkoxide (alkali glue) were synthesized in our lab [29]. Other reagents were used as received without further purification.

### 2.2. Characterization of Polymers

The instrument (Bruker Avance 400 MHz spectrometer, Mannheim, Germany) was used to record ^1^H NMR without tetramethylsilane as the internal standard, using deuterated chloroform (CDCl_3_) as solvent. The viscosity-average molecular weight (*M*_v_) was determined by the ubbelodhe viscometer (IVS-300, Zonmon Technology Co., Ltd., Hangzhou, China). Silicone oil was dissolved in chlorobenze0ne to prepare an approximate 4 mg/mL solution. The apparent viscosity was characterized by the rheometer (Brookfield R/S plus, Brookfield, WI, USA). An infrared spectrometer (Tensor37, Bruker, Mannheim, Germany) was employed in the range of 400–4000 cm^−1^ using potassium bromide (KBr) pellets prior to the measurements. Images of the SHSO-*g*-MBB were acquired on a Carl zeiss-Axio Scope.A1 microscope (POM) at a heating rate of 10 °C/min. The small angel X-ray scattering (SAXS) instrument (SAXSess mc2, Anton Paar, Graz, Austria) was employed for the measurements, equipped with a Kratky block-collimation system and a temperature control unit (Anton Paar TCS300).

### 2.3. Characterization of LCMVPQs and MVPQs

Cross-link density was confirmed using the VTMR20-010V-T magnetic resonance cross-link density spectrometer (Niumag Analytical Instrument Corporation, Suzhou, China).

The glass-transition temperature and phase-transition temperature were measured by DSC instrument (DSC1/700, Mettler-Toledo, Greifensee, Switzerland) in the atmosphere of nitrogen at a heating rate of 10 °C/min. The thermogravimetric (TG) data were collected on a Labsys Evolution TGA/DSC Synchronous Thermal Analyzer (Seteram, Lyon, France) from 30 °C to 800 °C at a heating rate of 10 °C/min in air. The mechanical properties of LCMVPQs and MVPQs were characterized by a universal tensile tester (Sansi Vertical Technology Co., Ltd., Shenzhen, China). The tensile strength and elongation at break were tested according to the GB/T528-2009 standard at a loading speed of 500 mm/min. The GT-GS-MB rubber hardness apparatus (Gotech Testing Machines Co., Ltd., Dongguan, China) was utilized to confirm the Shore A hardness. The DMA was performed in a tension mode with a DMA/DMTA (Electroforce 3200, Bose, USA) in the temperature range of −120 °C to 50 °C at a rate of 6 °C per minute and a constant frequency of 1 Hz. The specimen type was a rectangle with a width of 9 mm, a length of 20 mm and a thickness of 2 mm.

### 2.4. Synthesis of SHSO-g-MBB

SHSO-*g*-MBB was prepared using the thiol-ene “click” reaction of SHSO and MBB, using DMPA as the photoinitiator. We added SHSO, MBB, DMPA (0.1% *w*/*w*) and 20 mL toluene in a 50 mL round-bottomed flask with a mechanical stirrer and removed the air by bubbling dry nitrogen gas for 1 h. Next, the bottle was sealed and illuminated by 365 nm UV light for 20 min. Finally, the toluene was distilled under reduced pressure. Yield: 98%. ^1^H NMR (CDCL_3_, 400 MHz): δ 0.10 (s, –Si(CH_3_)_3_), 0.64 (m, –SiCH_2_CH_2_–), 1.36 (m, –SH), 1.60–2.00 (s, –SiCH_2_CH_2_CH_2_–, –SCH_2_CH_2_CH_2_), 2.48–2.77 (m, –CH_2_CH_2_S–, –CH_2_CH_2_SH), 3.81 (s, –OCH_3_), 4.05 (m, –CH_2_CH_2_O), 6.94, 7.11, 8.12 (d, –OC_6_H_4_COOC_6_H_4_OCH_3_).

### 2.5. Synthesis of Silicone Oils with Different Phenyl Content

Silicone oils with different phenyl content were synthesized by ring-opening equilibrium polymerization. Taking the synthesis of silicone oil with 40% phenyl content as an example, D_4_ (133 g), D_n_
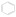
 (163 g), D_4_^vi^ (1 g) and MM^vi^ (1.140 mL) were added in a 500 mL, three-necked, round-bottomed flask with a mechanical stirrer. The mixture was placed in an oil bath at 50 °C under reduced pressure for about 1 h. After the water was removed, alkali glue (1% *w*/*w*) was added into the bottle. The reaction was carried out at 100 °C under nitrogen atmosphere. After 6 h, the system was heated to 150 °C for 0.5 h to decompose the alkali glue. In the end, low boiling components were eliminated by vacuum distillation at 200 °C. The product was a colorless and transparent silicone oil. Yield: 85%. ^1^H NMR (CDCL_3_, 400 MHz): δ 0.10 (s, –Si(CH_3_)_2_–), 0.33 (s, –SiC_6_H_5_CH_2_–), 5.76, 5.91, 6.10 (m, –Si(CH=CH_2_)–), 7.36, 7.59 (s, –SiC_6_H_5_CH_2_–). ^29^Si NMR (CDCL_3_, 400 MHz): δ –18, –21 (–Si(CH_3_)_2_O–), –32, –35 (–Si(C_6_H_5_)O–).

### 2.6. Fabrication of LCMVPQs and MVPQs

In a typical procedure, we dissolved SHSO-*g*-MBB in the DCM and transferred the solution to a round box. We let it stand for 2 h until the DCM evaporated completely. Silicone oils with different phenyl content, 0.1% DMPA as the photoinitiator and fumed silica H2000 were added into the above-mentioned box. Then, mixing was carried out five times in a three-dimensional high-speed mixer for 40 s each time. Finally, the mixture was placed onto the mould. After eliminating the air, it was illuminated by 365 nm ultraviolet light for 1 h. We removed the milky white film from the mold. The synthesis process of MVPQs was basically the same as that of LCMVPQs.

## 3. Results and Discussion

### 3.1. Synthesis and Characterization of SHSO-g-MBB, Silicone Oils with Different Phenyl Content, LCMVPQs and MVPQs

We prepared SHSO-*g*-MBB using the thiol-ene click reaction of SHSO and MBB [27,30]. Silicone oils with different phenyl content were synthesized by ring-opening equilibrium polymerization [31]. All the processes are illustrated in Figure 1.

The degree of polymerization (DP) of SHSO = 18/(I_5_ − 3 × I_1_), where I_5_/I_1_ is the ratio of integrated intensity of peak 5 to that of peak 1 in the ^1^H NMR spectrum of Figure 1a. The DP of SHSO was calculated to be ~10. It can be seen from Figure 1b that the grafting efficiency of MBB is almost 100% because the vinyl peak of MBB completely disappears. The thiol-ene click reaction is relatively convenient and efficient [32,33]. According to the following equation: the number of grafting MBB = 10 × (2 × I_5_/I_2_), I_5_ and I_2_ is the ratio of integrated intensity of peak 5 to that of peak 2 in the ^1^H NMR spectrum of Figure 1b, we can conclude that the number of grafting MBB was approximately six.

Taking the synthesis of silicone oil with 40% phenyl content as an example, the phenyl content = 3 × I_5_/(1.5 × I_5_ + (I_1 _+ I_2_)), I_5_, I_1_ and I_2_ is the integrated intensity of peak 5 and peak 2 in the ^1^H NMR spectrum of Figure 1c. The phenyl content of the obtained silicone oil is in accordance with the phenyl content of the feed (Figure 1c,d). The synthesis process of silicone oil in our research group is relatively mature and we can achieve the synthesis of silicone oil with different phenyl content [34,35].

We obtained five kinds of silicone oils with 0%, 5%, 10%, 20% and 40% phenyl content by ring-opening equilibrium polymerization to investigate the effect of phenyl content on the mechanical adaptability of LCMVPQs. The higher the phenyl content was, the worse the reactivity was. Therefore, we added moderate D4^vi^ in the preparation of silicone oils with high phenyl content to avoid the incomplete cross-linking caused by the decreased reactivity.

The composition and characterization of silicone oils are listed in Table 1. Additional ^1^H NMR spectrums of different phenyl content are available in Appendix A.

We prepared LCMVPQs or MVPQs using the thiol-ene click reaction of SHSO-*g*-MBB or SHSO and silicone oils with different phenyl content. The mechanical strength of the elastomers was improved by the physical blending of H2000 as a reinforcing filler.

All the synthesized LCMVPQs and MVPQs are summarized in Table 2. It can be seen from Table 2 that the difference of SHSO in the LCMVPQs or MVPQs is caused by the different reactivity of the silicone oils with different phenyl content.

In the FT-IR spectra, the absorption peaks near 1260 cm^−1^, 780 cm^−1^ are attributed to Si–Me, and the strong and wide absorption peak near 1080 cm^−1^, 1025 cm^−1^ is the stretching vibration absorption peak of Si–O–Si. The shear vibration absorption peak of the silicon–vinyl double bond is near 1410 cm^−1^. The vibration absorption peak of the silicon–phenyl aryl ring is near 1428 cm^−1^, and the stretching vibration absorption peak of the silicon–vinyl double bond is near 1610 cm^−1^. We can clearly see that the higher the phenyl content is, the more D_4_^vi^ we added, and the stretching vibration absorption peak of the silicon–vinyl double bond is more obvious (Figure 2a). The strong absorption peak near 2960 cm^−1^ is the stretching vibration absorption peak of methyl C–H, and the one near 3050 cm^−1^ is the stretching vibration absorption peak of H on the benzene ring skeleton. It can be seen in Figure 2a that with the increasing content of phenyl, the infrared peaks near 1428 cm^−1^ and 3050 cm^−1^ are more and more obvious. The appearance of the ester group (~1730 cm^−1^) indicates that the thiol-ene click reaction was successful and LCMVPQs and MVPQs were successfully prepared (Figure 2b).

### 3.2. Characterization of POM, SAXS and Rheological Properties

POM and SAXS were used to further explore the liquid crystal phase. Figure 3a,c show that the SHSO-*g*-MBB has nematic phases at room temperature. When the temperature is higher than the phase transformation temperature (*T*_NI_), the liquid crystal phase disappears under POM observations (Figure 3b), and the scattering peak of SAXS cannot be observed (Figure 3c). Furthermore, the rheological determination of SHSO-*g*-MBB further confirmed the *T*_NI_. As shown in Figure 3d, the storage modulus (G′) and loss modulus (G″) of SHSO-*g*-MBB decrease significantly when the temperature reaches 43 °C, and the damping factor increases from 0.2 to 1.3 at about 43 °C. The results indicate that the nematic-to-isotropic phase transformation temperature of the SHSO-*g*-MBB is approximately 43 °C.

### 3.3. Determination of Cross-Linking Density

In order to avoid the impact of cross-linking density differences on the mechanical properties of elastomers, we tried to ensure that the cross-linking density of all samples is basically the same. As presented in Table 3, the proportion of dangling chains of LCMVPQs is obviously higher than that of MVPQs, and the proportion of cross-linking chains of MVPQs is obviously higher than that of LCMVPQs. A possible explanation for this phenomenon is that the grafting of mesogenic units results in an increase in the dangling chains of elastomers, and the presence of the liquid crystal phase can hinder the thiol-ene click reaction. There is no obvious rule in the ratio of cross-linking and free chains in LCMVPQs and MVPQs, which is also related to the different reactivity of the silicone oils with different phenyl content.

### 3.4. The Thermal Stability and the Thermal Properties Characterization of All the LCMVPQs and MVPQs

The thermal stability of all the LCMVPQs and MVPQs was explored by TG analysis. All the LCMVPQs and MVPQs had good thermal stability and the thermal decomposition temperature was above 350 °C (Table 4). The 5%, 10% and maximum weight loss temperature of LCMVPQs 1–5 (Figure 4a) are lower than those of MVPQs 1–5 (Figure 4b), indicating that the introduction of mesogenic units reduced the thermal stability of LCMVPQs. This result can be explained by the fact that the mesogenic units are mainly composed of carbon–carbon bonds, whose thermal stability is not as good as that of the siloxane linkage. So, the grafting mesogenic units undergo thermal decomposition first [23]. In addition, due to the introduction of mesogenic units, the 800 °C residual weight percentages of LCMVPQs 1–5 are all less than that of MVPQs 1–5 (Table 4). The maximum weight loss temperature of LCMVPQs 1–5 or MVPQs 1–5 increases with the increase in phenyl content (Table 4). That is because phenyl is a large rigid group with a large steric hindrance that hinders the thermal degradation of the main chain polysiloxane [36,37]. The thermal behavior of LCMVPQs and MVPQs was examined by DSC. We can observe from Figure 4c that the *T*_NI_ of the SHSO-g-MBB is roughly 45 °C, which is basically consistent with previous POM, SAXS and rheological results. The *T*_NI_ peaks of LCMVPQs are all observed near 45 °C and no *T*_NI_ peak is found in MVPQs. This shows that the change of phenyl content does not affect the thermotropic phase transition of LCMVPQs. It is obvious that the glass-transition temperature (*T*g) of LCMVPQs and MVPQs gradually increased from −125 °C to −77 °C with the increase in phenyl content. Dimethyl silicone rubber crystallizes at about −100 °C, the melting temperatures (*T*m) is approximately −40 °C. The *T*m peak disappears in LCMVPQ 2–5 and MVPQ 2–5. As expected, the introduction of the phenyl group destroyed the regularity of dimethylsiloxane and affected the crystallization of the silicone rubber (Figure 4c,d).

### 3.5. Stress-Strain Behavior

The effect of phenyl content on the mechanical adaptability of LCMVPQs and MVPQs was investigated using a universal tensile tester. In previous work, the mesogenic units were arranged along the direction of the applied stress during the stretching process, namely the mechanical adaptability. To provide an insight into the effect of phenyl content on mechanical adaptability, we prepared MVPQs with different phenyl content without a liquid crystal group for comparison (Figure 5b). It is obvious that the stress-strain curve trends of LCMVPQ 4 and 5 are roughly the same as those of MVPQ 4 and 5, which indicates that the mechanical adaptability basically disappears when the phenyl content is more than 20%. It can be seen from Figure 5a that the mechanical strength of LCMVPQ 1–3 is lower than that of MVPQ 1–3 and the elongation at break of LCMVPQ 1–3 is greater than that of the control group. Compared with MVPQs, the mechanical adaptability of LCMVPQ 1–3 became less and less obvious with the increase in phenyl content (Figure 5a,b). A possible explanation for this might be that with the increase in phenyl content, phenyl will hinder the rearrangement of the mesogenic units along the applied stress direction. The rigid groups with large steric hindrance can prevent the reversible phase transition of the liquid crystal elements induced by the stress, and the mechanical adaptability gradually disappears. The details are shown in Table 5.

### 3.6. Dynamic Thermomechanical Analysis (DMA)

The viscoelastic property of LCMVPQs and MVPQs was evaluated by DMA. It can be concluded from Figure 6 that the *T*g of the LCMVPQs and MVPQs increases from −122 °C to −74 °C with the increase in phenyl content, pure dimethyl silicone rubber crystallizes at about −100 °C and the *T*m is approximately −40 °C, which is consistent with DSC results. With the increase in phenyl content, the loss modulus of LCMVPQs or MVPQs becomes greater (Figure 6c,d). This indicates that the internal friction of LCMVPQs or MVPQs’ molecular chain movement becomes greater. The loss factor of LCMVPQs increases from 1.12 to 1.32, and the loss factor of MVPQs increases from 1.35 to 1.76 (Figure 6e,f). Hence, with the increase in phenyl content, the damping performance becomes better. It must be pointed out that the loss factors of LCMVPQ 1 to 5 are all less than those of MVPQ 1 to 5. The introduction of the liquid crystal phase impeded the internal friction of the LCMVPQs’ molecular chain movement and reduced the damping performance of silicone rubber, which has not yet been reported. Introducing the liquid crystal phase into silicone rubber is also a very effective method to control the damping properties of silicone rubber.

## 4. Conclusions

In conclusion, we successfully prepared liquid crystal-based phenyl silicone rubbers with different phenyl content by two-step successive click reactions and systematically explored the effect of phenyl content on their mechanical adaptability. The changes in phenyl content did not affect the thermotropic phase transition of LCMVPQs. With the increase in phenyl content, the steric hindrance of phenyl can hinder the rearrangement of the mesogenic units along the applied stress direction. This property can help us to prepare elastomers with different mechanical adaptive characteristics for different applications. Furthermore, it is the first time that we found that the liquid crystal phase in the LCMVPQs impedes the internal friction of molecular chain movement. This work achieves the regulation of mechanical adaptability, which is of great significance to the modification and theoretical exploration of silicone rubber and is expected to be applied in practical application fields.

## Data Availability

Not applicable.

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
