# Peer review of "The Effect of Phenyl Content on the Liquid Crystal-Based Organosilicone Elastomers with Mechanical Adaptability"

_polymers, 2022, doi:10.3390/polym14050903_

Round 1

Reviewer 1 Report

The manuscript is interesting and can be process further; however, I have following Questions

  • The authors are required to mention the presented work application in detail.
  • Editorial, Typos were found in the manuscript and particularly English is very poor.
  • Follow the journal reference style.
  • Figure 1 is blurred, it is required to redrawn to make it more clear.
  • It seems that text are copied and highlighted, for example (table 2, caption of Figure 1 and many more). Please modify it.
  • Rephrase the heading of section 3.4.
  • The results of TGA in Fig. 3(a) and (b) does not very much. It is required to give its significance.
  • The repetition abstract and conclusion should be avoided.

Reviewer 2 Report

The manuscript addressed to study the effect of phenyl Content on the liquid crystal-based organoganosilicone elastomers with supreme mechanical addaptabillity. However, why the authors refer as "supreme" mechanical adaptability. In this regard, a bibliographical support in results section is needed to back this characteristic that it is only mentioned in title and introduction section. 

Round 2

Reviewer 1 Report

The revision is not adequate.

Author Response

I am very grateful to your comments for the manuscript. We have made revision related to the English language of the article. All the modifications were marked in red color. We hope that the correction will meet with your approval.

Reviewer 2 Report

This reviewer does not have any other comment, and recomend accept in the present form the manuscript. 

Author Response

感谢您对我们工作的支持。